# Integrative Analysis of Gene Expression and Promoter Methylation to Differentiate High-Grade Serous Ovarian Cancer from Benign Tumors

**DOI:** 10.3390/biomedicines13020441

**Published:** 2025-02-11

**Authors:** Ieva Vaicekauskaitė, Paulina Kazlauskaitė, Rugilė Gineikaitė, Rūta Čiurlienė, Juozas Rimantas Lazutka, Rasa Sabaliauskaitė

**Affiliations:** 1Institute of Biosciences, Life Sciences Center, Vilnius University, 10257 Vilnius, Lithuania; 2National Cancer Institute, 08406 Vilnius, Lithuania; 3Institute of Biomedical Sciences, Faculty of Medicine, Vilnius University, 08661 Vilnius, Lithuania; 4National Cancer Center, Vilnius University Hospital Santaros Klinikos, 08661 Vilnius, Lithuania

**Keywords:** high-grade serous ovarian carcinoma, gene expression, gene promoter methylation, diagnostic model

## Abstract

**Background:** Ovarian cancer (OC) is the third most common and second most lethal onco-gynecological disease in the world, with high-grade serous ovarian cancer (HGSOC) making up the majority of OC cases worldwide. The current serological biomarkers used for OC diagnosis are lacking sensitivity and specificity, thus new biomarkers are greatly needed. Recently, the chromatin remodeling complex gene *ARID1A*, Notch and Wnt pathway gene expression, as well as *HOX*-related gene promoter methylation have been linked with promoting OC. **Methods:** In this pilot study, 10 gene expression biomarkers and 4 promoter methylation biomarkers were examined as potential diagnostic and prognostic indicators of OC in 65 fresh-frozen gynecologic tumor tissues. **Results:** Out of 10 genes analyzed, the expression of eight biomarkers was significantly reduced in OC cases compared to benign, and *HOX*-related gene promoter methylation significantly increased in OC tumors. Out of 14 biomarkers, *CTNNB1* showed the best single biomarker separation of HGSOC from benign cases (AUC = 0.97), while a combination of the seven Notch pathway-related gene expressions (*NOTCH1*, *NOTCH2*, *NOTCH3*, *NOTCH4*, *DLL1*, *JAG2*, and *HES1*) demonstrated the best separation of HGSOC from the benign cases (AUC = 1). **Conclusions:** The combination of multiple gene expression or gene promoter methylation biomarkers shows great promise for the development of an effective biomarker-based diagnostic approach for OC.

## 1. Introduction

Ovarian cancer (OC) is the second leading cause of death due to onco-gynecological disease, with more than 300 thousand new cases and 200 thousand deaths reported worldwide each year [1]. The vast majority (90%) of OC cases are of the epithelial type OC, with the largest share (52%) of cases further categorized into high-grade serous type OC (HGSOC). Compared to other types of epithelial OC (endometrioid, clear cell, mucinous, borderline) or non-epithelial (stromal and germ cell) tumors, HGSOC is diagnosed at a late stage (80% of HGSOC diagnosed at stage III-IV, compared to ~60% of other epithelial OC diagnosed at stage I) and is responsible for the largest share of fatalities, with a 5-year survival rate of only 43%, compared to 66–82% for other types of epithelial OC [2]. Despite the characteristically low survival rates (5-year age-standardized net survival rate in Lithuania and the majority of Europe as low as 30–40%) slowly trending upwards [3], it remains critical to have great diagnostic and prognostic tools for OC.

OC does not present any specific symptoms; the only physical indication of a possible tumor are abdominal distention and pain, which are symptoms more typically accredited to gastrointestinal problems. The main diagnostic tests performed for OC are abdominopelvic ultrasonography and the measurement serum protein biomarker CA125 [4]. Despite the wide use of CA125, it is recommended against OC screening using CA125, due to the CA125 blood test failure to reduce the mortality of OC patients [5]. The attempts to increase the accuracy of CA125 with the addition of another serum biomarker HE4 do not provide the desired specificity for a screening assay (sensitivity 79%, specificity 91%) [6].

The current goal for OC researchers is to find new molecular characteristics that could detect cancer and identify patients suitable for targeted treatment [7]. The Cancer Genome Atlas (TCGA) project has been a major effort in identifying genetic alterations prevalent in cancers. It found that approximately 22% of HGSOC patient samples exhibit dysregulation in the Notch pathway [8]. In contrast, the Wnt pathway and the chromatin remodeling complex gene *ARID1A* are among some of the most dysregulated in non-serous types of OC [9]. Studies show that Notch and Wnt pathways are important regulators of the stemness, differentiation and proliferation in the fallopian tube cells where HGSOC is thought to originate [10], thus its expression levels have the potential to be used as early biomarkers of HGSOC. *CTNNB1* and *FBXW7* genes are some of the most altered Wnt pathway genes in OC [9], while in the Notch pathway the receptors (*NOTCH1-4*), ligands (such as *DLL1*, *JAG2*), as well as the canonical target (*HES1*), are some of the most dysregulated Notch genes in cancers overall [11].

Chromatin remodeling complexes contribute to non-gene specific regulation of transcription via control of DNA accessibility to the transcription factors. The process of transcription repression is kickstarted by DNA methyltransferase (DNMT) methylating the DNA cytosine base in the CpG-rich dinucleotide sequences known as CpG islands, frequent around most gene promoters. DNA methylation facilitates the recruitment of histone deacetylases that result in more condensed and transcriptionally inactive chromatin [7]. The hypermethylation of the tumor suppressor gene promoter CpG islands is one of the earliest cancer hallmarks, often predating genetic changes and leading to altered gene expression. Compared to normal fallopian tube epithelium, at least 3868 genes in HGSOC samples are hypermethylated and affect gene transcription [12].

In many types of cancer, homeobox (*HOX*) genes recruit PRC2 repressive chromatin regulation complex and thus act as transcriptional regulators [13]. *HOX* genes are often regulated via the hypermethylation of CpG islands in their promoter regions. *CDX2*, *ALX4* and *HOPX*, three non-canonical *HOX* genes are hypermethylated in colorectal, pancreatic, gastric and lung cancers [12]; however, the data on their methylation status in OC is less clear. Although the *ARID1A* gene is the second most mutated gene in non-HGSOC OC, its loss in gynecologic tumor precursors such as endometriosis could be caused by promoter hypermethylation as well [14].

Importantly, the hypermethylation of *HOX*-related genes and the loss of functional *ARID1A* are highly associated with resistance to chemotherapy and are potentially druggable through epigenetic therapies [13,15], with DNMT inhibition therapies potentially effective in resensitizing *HOX* deficient tumors to carboplatin [16], and recent clinical trials running for EZH and BET inhibitors for *ARID1A* deficient tumors [17].

As part of the goal of creating a biomarker-based diagnostic test for OC, this pilot study examined nine main gene expression biomarkers of the Notch and Wnt pathways and the OC susceptibility gene *ARID1A* for expression changes in patients with gynecologic tumors. In addition, gynecologic tumor tissue was analyzed for promoter methylation of *ARID1A* and three homeobox (*HOX*) related genes. Multiple biomarkers incorporated into the model outperformed single biomarkers in the separation of HGSOC disease from benign counterparts and other types of gynecologic disease, thus providing a promising strategy for the further development of potential diagnostic tests for the detection of HGSOC.

## 2. Materials and Methods

### 2.1. Patient Selection and Sample Collection

The study cohort comprised of 65 patients with suspected ovarian cancer who received salpingo-oophorectomy at the National Cancer Institute, Lithuania, between 2018 and 2023. The study approval was granted by the regional bioethics committee (No. 158200-18/5-988-539). Tissue samples collected from the ovary tissues were removed during the procedure, flash-frozen and immediately stored at –80 °C until further use.

Of the 65 patients included in the study, 56 had malignant gynecologic tumors, 42 had confirmed HGSOC, and 14 had other gynecologic tumors (of them five cases were of borderline ovarian cancer, three synchronous endometrial and endometrioid ovarian cancer, two mucinous, one low grade serous, one endometrioid, one clear cell ovarian cancer, and one granulosa cell tumor), while nine had benign gynecologic conditions (gynecologic cysts or benign tumors, as well as one case of preventative salpingoovarectomy due to *BRCA2* mutation). The main clinical features in the study included age at diagnosis, tumor grade, International Federation of Gynecology and Obstetrics (FIGO) stage, and pre-operation serum CA125 concentration, where 35 U/mL was considered a threshold concentration and values below 35 U/mL were considered normal [4]. The clinical features of the patients are provided in Table 1.

### 2.2. Nucleic Acid Extraction

A small tumor tissue sample (~30 mg) was pulverized in liquid nitrogen using a mortar and pestle and a small amount (~10 mg) was used for either RNA or DNA extractions. DNA samples were extracted first by digesting the samples with Proteinase K (Thermo Scientific, Thermo Fisher scientific (TFS), Vilnius, Lithuania) for 16 h and then performing a standard phenol-chloroform DNA purification and ethanol-based precipitation method. This DNA extraction method is one of the longest-established methods widely used for high-purity and high-molecular-weight DNA extraction and is highly suitable for multiple applications [18]. RNA from ovarian tissues was extracted using a widely validated method using TRIzol Reagent (Invitrogen, TFS, Carlsbad, CA, USA) [19]. In addition to its effective tissue lysis, TRIzol Reagent preserves RNA integrity and stability, ensuring high-quality total RNA [20] suitable for downstream applications, such as cDNA synthesis, delivering consistent results. The purity and concentration of nucleic acid extractions were evaluated using a Nanodrop 2000 spectrophotometer (Thermo Scientific, Wilmington, DE, USA). The nucleic acid samples were stored at –80 °C until use in the methylation-specific PCR (MSP) or cDNA synthesis step.

### 2.3. Synthesis of cDNA and Quantitative PCR

RNA extracted from gynecologic tissues was used for quantitative analysis of gene expression. A two-step RT-qPCR procedure was followed: 0.5 µg of the total RNA was used for cDNA synthesis following the manufacturer’s protocols of the Maxima First Strand cDNA Synthesis Kit for RT-qPCR with dsDNase (Thermo Scientific, TFS, Vilnius, Lithuania) and ProFlex PCR System (Applied Biosystems, TFS, Singapore). The cDNA synthesis kit produces a high yield of RNA and is highly suitable for subsequent PCR applications [21]. Quantitative PCR (qPCR) was performed on a QuantStudio 5 Real-Time PCR System (Applied Biosystems, TFS, Singapore) using Maxima SYBR Green qPCR Master Mix (2X) kit (Thermo Scientific, TFS, Vilnius, Lithuania) in accordance to the manufacturer’s instructions.

A 10-gene panel consisting of Notch pathway genes (*NOTCH1*, *NOTCH2*, *NOTCH3*, *NOTCH4*, *JAG2*, *DLL1*, *HES1*), Wnt pathway genes (*CTNNB1*, *FBXW7*) and a chromatin remodeling complex SWI/SNF gene *ARID1A* was analyzed for changes in relative gene expression in gynecologic tissues. *GAPDH* was used as an internal control gene. The primer sequences used in qPCR reactions are provided in Appendix A. An initial analysis of Ct values was conducted on QuantStudio Design and Analysis Software v1.4.3 (Applied Biosystems, TFS, Singapore). Gene expression normalization was performed using the log_2_ 2^−ΔCt^ method; these values were used in statistical analysis.

### 2.4. Methylation Specific PCR

*ARID1A* and three *HOX*-related gene (*HOPX*, *ALX4*, *CDX2*) promoter methylation status were tested using methylation-specific PCR (MSP). First, bisulfite modification was applied to gynecologic tissue DNA using an EZ DNA Methylation kit (Zymo Research, Irvine, CA, USA). The MSP primers used in the analysis are provided in Appendix A. MSP assays were performed according to Phusion U Hot Start DNA Polymerase (Thermo Scientific, TFS, Vilnius, Lithuania) protocol instructions. According to the three-step cycling protocol the annealing step consisted of 30 s at 60 °C for *ALX4* and *ARID1A*, 58 °C for *CDX2*, and 62 °C for *HOPX*. All reactions were performed on the ProFlex PCR System (Applied Biosystems, TFS, Singapore). For each set of reactions, CpG Methylated Human Genomic DNA (Thermo Scientific, TFS, Vilnius, Lithuania) was a positive control and a mix of five healthy human genomic DNA as a wild-type control as well as nuclease-free water for non-template control. The MSP products were visualized using 3% agarose gels (Carl Roth, Karlsruhe, Germany) in 1× TAE buffer (Thermo Scientific, TFS, Vilnius, Lithuania) with 0.6 µg/mL Ethidium Bromide (Carl Roth, Karlsruhe, Germany) staining. TriTrack DNA loading dye (6×) (Thermo Scientific, TFS, Vilnius, Lithuania) was used for loading the gels, and the GeneRuler 100 bp DNA ladder (Thermo Scientific, TFS, Vilnius, Lithuania) was used as a size standard. Gels visualized using UVP ChemStudio (Analytic Jena, Jena, Germany) imaging system.

### 2.5. Statistical Analysis

For comparisons of two independent groups, Welch’s *t*-test [22] was chosen due to its high statistical power and ability to minimize the risk of invalid statistical inferences due to unequal sample sizes [23]. The Benjamini-Hochberg method was applied to adjust *p*-values to effectively control false discovery rates [24]. The contingency between methylation biomarkers and categorical variables examined using Fisher’s exact test due to the small sample size. Receiver operating characteristic (ROC) analysis was used to analyze the performance of biomarkers or their combinations, while the DeLong test was used to compare ROC curves. Logistic regression models were applied for combinations of multiple biomarkers. All analysis and visualization were performed on R (version 4.3.1, R Foundation for statistical computing, Vienna, Austria). The ComplexHeatmap package (version 2.16.0) was used for heatmap visualizations, the pROC package (version 1.18.5) for ROC analysis, the glmnet package (version 4.1-8) for regression analysis and the ggplot2 package (version 3.5.1) for general visualizations. The results were considered statistically significant when *p* < 0.05.

## 3. Results

### 3.1. Promoter Methylation Changes in Ovarian Cancer Compared to Benign Gynecologic Tissues

We analyzed 65 gynecologic tumor tissue samples using a panel composed of four gene promoter methylation biomarkers and 10 gynecologic cancer-related mRNA expression biomarkers. Promoter methylation of at least one *HOX* related gene (*ALX4*, *CDX2* or *HOPX*) or *ARID1A* gene was detected in 83.08% (54/65) of tumor samples. *ALX4* promoter methylation was detected in 44.62% of samples, *CDX2* 49.23%, *HOPX* 33.85% and *ARID1A* 43.08% (Figure 1).

Comparing gene promoter methylation in OC cases with benign tissues, all three *HOX*-related gene promoters had significant increases (*p* < 0.04) in methylation frequency in OC; with both *ALX4* and *CDX2* only 1/9 benign cases showed promoter methylation and no benign cases had *HOPX* promoter methylation. *ARID1A* promoter methylation showed no significant difference in OC and benign cases (Appendix A).

When compared to benign cases, the significant increase in promoter methylation in HGSOC was detected only for the *HOPX* promoter (*p* = 0.04); however, a tendential increase (*p* = 0.06) was also noticed in *ALX4* and *CDX2* gene promoters. When comparing the other OC cases (non-HGSOC OC) to benign counterparts, *CDX2* and *HOPX* showed increased promoter methylation (*p* = 0.01 and 0.05, respectively) (Figure 2). Additionally, in the HGSOC group, a significant increase in *ARID1A* gene promoter methylation was found in advanced FIGO stage cases (*p* = 0.05) (Appendix A). No further associationsbetween gene promoter methylation status and clinical features of serum CA125 concentration, FIGO stage, grade or age were found (Figure 1).

### 3.2. Gene Expression Changes in Ovarian Cancer Compared to Benign Gynecologic Tissues

The expression levels of the seven Notch pathway, two Wnt pathway (*CTNNB1* and *FBXW7*), and *ARID1A* genes were also analyzed in ovarian tissue samples; 8/10 gene expression was significantly lower in OC cases compared to benign (*p* < 0.03, Fold change (FC) > 1.18), with only *ARID1A* showing a tendential reduction in expression (*p* = 0.06, FC = 1.14); *JAG2* was not significantly changed (Appendix A).

Comparing HGSOC samples with benign samples, a significant reduction in all but *JAG2*, *NOTCH3* and *ARID1A* gene expression was noted, with *CTNNB1* expression decreasing by two-fold (FC = 2.01; *p* < 0.001), which was the most out of all genes included in the panel. Notably, reduced *DLL1* and *HES1* expression was also detected in HGSOC when compared to other (non-type II) gynecologic tumors (FC = 1.17 and 1.43, respectively; *p* < 0.02). Notch receptor *NOTCH2* and Wnt gene *CTNNB1* expression were also significantly reduced in other gynecologic tumors when compared to benign tissues (FC = 1.32, and 1.62, respectively; *p* = 0.002) (Figure 3).

*HES1* and *CTNNB1* expression was significantly lower in high-grade (grade 3) gynecologic cancer compared to low-grade (grade 1) cases (*p* < 0.05, FC= 1.48, 1.22, respectively), while *DLL1* and *FBXW7* also showed a tendency for reduced expression in high-grade tumors (*p* = 0.05, FC = 1.17 and 1.13, respectively) (Appendix A). In the HGSOC group, Wnt gene *CTNNB1* and *FBXW7* expression was significantly correlated with advanced FIGO stage (IV vs. III, FC = 1.20, *p* = 0.04, FC = 1.13, *p* = 0.01, respectively) (Appendix A).

Wnt genes *CTNNB1*, *FBXW7*, and Notch genes *NOTCH2*, *NOTCH3*, and *NOTCH4* were significantly decreased in cases with a positive CA125 status (CA125 serum concentration > 35 U/mL, cases with increased CA125 (n = 50), CA125 normal serum concentration (n = 5), FC = 1.59, 1.20, 1.29, 1.28, and 1.33, respectively, *p* < 0.05) (Appendix A). No gene expression correlation with age at diagnosis was detected (Figure 1).

### 3.3. Building an Ovarian Cancer Diagnostic Model

Fourteen biomarkers were analyzed for their prediction of ovarian cancer using ROC analysis. The best predictor of OC vs. benign gynecologic tumors was *CTNNB1* expression (AUC = 0.95, accuracy = 0.86, sensitivity = 0.84, specificity = 1); however, the highest accuracy and sensitivity were detected by *NOTCH2* (AUC = 0.89, accuracy = 0.89, sensitivity = 0.89, specificity = 0.89) (Appendix A). In the promoter methylation biomarker group, the best separator of OC was *CDX2* (AUC = 0.72, accuracy = 0.60, sensitivity = 0.55, specificity = 0.89), which was marginally better than other *HOX* family gene promoters and a significant increase in AUC over *ARID1A* promoter methylation (*ARID1A* promoter methylation AUC = 0.49, DeLong test *p* = 0.03) (Appendix A).

The best single biomarker separator of HGSOC from the benign tumor group was *CTNNB1* gene expression (AUC = 0.97). In general, gene expression biomarkers were better suited to detect HGSOC than promoter methylation biomarkers with with 6/10 (*CTNNB1*, *DLL1*, *HES1*, *FBXW7*, *NOTCH2*, *NOTCH4*) of gene expression biomarkers significantly improving HGSOC separation from benign samples compared to best promoter methylation separators *CDX2* and *ALX4* (AUC = 0.69, DeLong *p* < 0.03) (Figure 4).

We then examined biomarker combinations for their ability to separate the OC cases from benign gynecologic tumors. All five combinations were tested, combining all 14 available biomarkers, i.e., 10 gene expression and 4 gene promoter methylation (mixed biomarker combination), all gene expression biomarkers (10 biomarkers), all promoter methylation biomarkers (four biomarkers); the Notch gene expression biomarker combination which included seven Notch genes in our panel (*NOTCH1*, *NOTCH2*, *NOTCH3*, *NOTCH4*, *DLL1*, *JAG2*, and *HES1*), and the three *HOX*-related gene promoter methylation combination. Comparing the combinations we found that a 10 gene expression biomarker combination was able to perfectly separate the two groups. While all five combinations significantly outperformed serum CA125 in both AUC and specificity (CA125 AUC = 0.78, specificity = 0.57), only the 14 biomarker, 10 gene expression biomarker combination and Notch gene expression biomarker combination exceeded CA125 sensitivity and accuracy (sensitivity = 0.98, accuracy = 0.93) and significantly improved on the OC separation form benign samples (*p* < 0.05) (Appendix A).

To achieve the perfect separation of HGSOC cases from benign gynecologic tumors (AUC = 1) the combination of only the seven Notch pathway gene expression was sufficient. While the combination of biomarkers did not significantly improve the separation compared to the best single biomarker *CTNNB1* (*p* = 0.13), it did improve the accuracy and sensitivity (1.00 and 1.00 for combination and 0.92 and 0.91 for *CTNNB1* expression, respectively). The promoter methylation biomarker models including the three *HOX*-related genes or all four methylation biomarkers outperformed serum CA125 specificity, but only the gene expression combinations outperformed HGSOC separation from benign cases (*p* = 0.04) (Figure 5). Overall, the combination of biomarkers improved both serum biomarker and gene promoter methylation separation of OC or HGSOC samples from benign cases and improved on specificity and accuracy of the potential biomarker test when compared to single biomarkers.

## 4. Discussion

The present study investigated 10 gene expression and four gene promoter methylation biomarkers in gynecologic tumor tissue for the potential of ovarian cancer diagnosis. The biomarkers comprised a set of seven Notch and two Wnt pathway gene expressions, alongside gene expression and promoter methylation of the chromatin remodeling complex SWI/SNF gene *ARID1A*, as well as three *HOX*-related gene promoter methylation analyses. To our best knowledge, this is the first such type of study that approaches a novel multivariable OC diagnosis based on different biomarker combinations.

Notch and Wnt are major cell signaling pathways that have been long associated with gynecologic tumors, both with roles in stemness, as evidenced by the fallopian tube organoids [25]. Notch pathway negatively impacts the activity of the Wnt pathway in most cancers; Wnt can also impact the expression of DLL1, HES1 and NOTCH2 on the protein level, while β-catenin can also interact with NOTCH1 and reduce its ubiquitination [26]. A recent analysis of the cancer genome atlas (TCGA) ovarian cancer data, largely comprised of HGSOC samples, has shown a significant reduction in *DLL1*, *JAG1*, *NOCTH4*, and an increase in *HES1*, *NOCTH1* and *NOTCH3* expression in OC samples compared to normal tissues [27]. The reduction in gene expression was consistent with our results of lower Notch gene expression in HGSOC samples compared to benign tumor tissues. The TCGA study also revealed significant associations between *JAG2* and *NOTCH1* and progress-free survival and *NOCTH4* with the FIGO stage, the latter was not detected in our study [27]. A similar study did not find any significant changes in *CTNNB1* mRNA expression in HGSOC compared to normal tissues (The Genotype-Tissue Expression project (GTEx)); however, there was a correlation between *CTNNB1* expression and clinical stage, also confirmed in our study [28]. Meanwhile, *FBXW7* has been found to be downregulated in ovarian cancer cells in TCGA data, tumor tissues and cell lines as well [29].

Homeobox (*HOX*) genes are an evolutionary conservative group of transcription factors bearing a homeodomain that have important functions in embryogenesis and carcinogenesis. Many *HOX* genes are described as tumor-suppressors and are downregulated via promoter hypermethylation [13]. The homeobox gene superclass has 235 functional genes, *CDX2* belongs to the ANTP-class, the most well-known class of *HOX*, while *ALX4* and *HOPX* belong to PRD-class [30], showing a vast diversity in the *HOX* genes.

*HOPX* expression is a novel tumor-associated stromal biomarker, it was found to be a potential molecular driver of OC, particularly of the “mesenchymal” OC subtype [31]. *HOPX* has been identified as one of the most cancer-specific hypermethylated biomarkers, with hypermethylation detected in 90% of primary gastric tumors [32], 62.5% (10/16) colorectal [33], and 69% (9/14) endometrial cancers [34], the latter was considerably higher than the 39% hypermethylation in our OC samples.

*ALX4*, another tumor suppressor in the *HOX* family is involved in the promotion of ovarian cancer metastasis: ALX4, together with the other HOX protein HOXB13 promotes epithelial to mesenchymal transition factor SLUG in ovarian cancer cell lines [35]. Moreover, TCGA analysis via weighted gene co-expression network analysis (WGCNA) selected *ALX4* as one of the 14 differentially expressed genes associated with ovarian cancer overall survival [36]. According to TCGA and GEO data, *ALX4* is also amongst the most differentially methylated genes in ovarian cancer, and its hypermethylation is associated with chemoresistance [37]. The *ALX4* promoter was also methylated in 69.44% (75/108) of cases and none of the eight normal breast tissues [38] while in colorectal cancer patients’ serum, methylation was detected in 68% (17/25) of cases and 12% (3/25) of controls [39], a similar rate to our 50% and 11% in OC and benign cases, respectively.

*CDX2* promoter methylation was detected in 71.67% (43/60) of colorectal cancer cases [40], and 55.45% (61/110) of lung cancer cases [41]. Interestingly, CDX2 was found to inhibit Wnt signaling in lung cancer cells via suppression of c-MYC, cyclin D1 and survivin expression [41]. Importantly, β-catenin and CDX2 nuclear expression have been associated with longer survival [36]. In another ovarian cancer study, immunohistochemical staining showed that only 15.1% (8/53) of tissues show *CDX2* expression and CDX2 was found to contribute to chemoresistance via the promotion of *MDR1* (multidrug resistance 1) gene expression [42].

TCGA comparison to GTEx normal tissue gene expression of Notch pathway genes showed an AUC of *NOCTH1* (0.603), *NOTCH2* (0.532), *NOTCH3* (0.739), *NOTCH4* (0.948), *HES1* (0.642), *DLL1* (0.823), *JAG2* (0.769) [27], while in comparison our smaller study showed slightly higher AUC values of *NOTCH1* (0.765), *NOTCH2* (0.884), *HES1* (0.950), *DLL1* (0.907), and lower AUCs of *NOCTH3* (0.714), *NOTCH4* (0.907) and *JAG2* (0.500) when comparing HGSOC and benign gynecologic tumors.

A few other publications have previously hypothesized that Notch pathway genes correlate with ovarian cancer recurrence and can discriminate between high- and low-risk OC. A 10 Notch gene combination, which included *HES1* and *JAG2*, has been able to classify TCGA and two other OC datasets into high- and low-risk groups independently from other clinical variables [43]. Another 11 Notch genes successfully differentiated OC from normal tissues and its signature predicted a 10-year prognosis with an AUC of 0.79 [44]. On the other hand, gene promoter methylation combination models have also been suggested for discrimination of HGSOC vs. benign tissues, with *HOXA9* (another *HOX* family gene), *EN1* and CA125 pre-operative levels reaching a sensitivity of 98.8% and specificity of 91.7%, while single *HOXA9* had a comparatively lower sensitivity of 95% [45].

While OC is commonly stratified by tumor histology into Type I and Type II disease, it is clear that genetic and epigenetic variation exists even within histological groups. Even within the Type II (HGSOC) group, which is believed to almost unanimously present with *TP53* mutations, there is significant heterogeneity in the prognosis and molecular features, specifically in gene expression. This variability was first identified by the initial TCGA ovarian cancer study and has been confirmed by many others. However, to this day, no genetic test or specific treatment has been developed to account for these variations [46]. In our study, the *CTNNB1*, *FBXW7*, *NOTCH2* mRNA as well as the *ARID1A* promoter hypermethylation biomarkers were associated with the HGSOC tumor stage, suggesting possible implications for the prognosis, despite our study lacking more in-depth prognosis data. The identification of new OC biomarkers could also inform clinical decisions and new treatment options. Chemoresistance is one of the hallmarks of HGSOC; as highlighted above, *ALX4*, and *CDX2* promoter methylation biomarkers are associated with chemoresistance; an early study in colon cancer cell lines showed successful restoration of *CDX2* expression via epigenetic therapies, such as DNMT inhibitors [47], thus proving a novel treatment approach for *HOX*-related gene expression deficient tumors. Similarly, *ARID1A* loss is associated with poor prognosis and is targetable via EZH2, HDAC6, multikinase or PARP inhibition [48].

The proposed diagnostic panels were able to improve on the current biomarker CA125 separation of the malignant tumors and most strikingly outperformed CA125 in specificity by 43% (CA125 specificity 0.57, gene expression combination specificity = 1). CA125 serum concentration increase may be present in cases of non-gynecologic conditions, such as osteoporosis, osteoarthritis, and hypercholesterolemia, causing the CA125 specificity to fall to 73–77% [49]. Low specificity is one of the downfalls of the CA125 and other serum biomarker tests as it may result in high false positive rates and cause unnecessary treatment through invasive tests, such as biopsies, thus biomarkers with lower specificity rates are not suitable for screening purposes. The adoption of more precise genetic biomarker-based tests could improve unnecessary treatments and may be an option for OC screening.

Although our study provided a relatively small cohort of ovarian cancer cases, the Notch gene biomarker combination was able to perfectly discriminate HGSOC cases from the benign gynecologic tumor cases, albeit the best single biomarker *CTNNB1* was also able to sufficiently separate these cases by itself. The inclusion of multiple biomarkers is more likely to address the complexity and heterogeneity of HGSOC, shown by the improved diagnostic accuracy and sensitivity compared to single biomarkers (a 9% improvement in sensitivity, and 8% improvement in accuracy in the gene expression biomarker model compared to *CTNNB1* expression alone). However, a larger study of non-invasive samples is necessary to confirm if the proposed biomarker panels can reach the necessary specificity of 99.6% and sensitivity of 75% needed for screening tests [50].

## 5. Conclusions

The present study shows the utility of epigenetic and transcriptional biomarkers and their combinations in the discrimination of HGSOC from benign or non-serous gynecologic tumors. By applying logistic regression, we determined that a combination of transcriptomic and epigenetic biomarkers exhibit more diagnostic accuracy than single biomarkers alone. While further validation in larger cohorts and liquid biopsy samples are necessary for developing a feasible screening strategy for OC, the study provides the groundwork for building multiomic diagnostic approaches for OC.

## Figures and Tables

**Figure 1 biomedicines-13-00441-f001:**
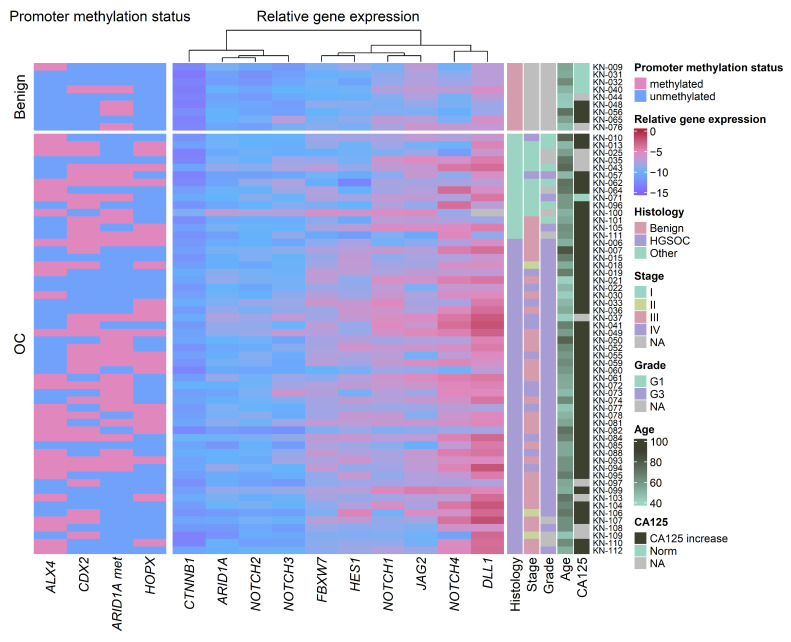
A heatmap of promoter methylation and relative gene expression in gynecologic tumors. *ARID1A met*—*ARID1A* promoter methylation, OC—ovarian cancer, HGSOC—high-grade serous ovarian cancer, other—other malignant gynecologic tumors, G1—grade 1, G3—grade 3, CA125 increase—CA125 serum concentration > 35 U/mL, NA—data not available.

**Figure 2 biomedicines-13-00441-f002:**
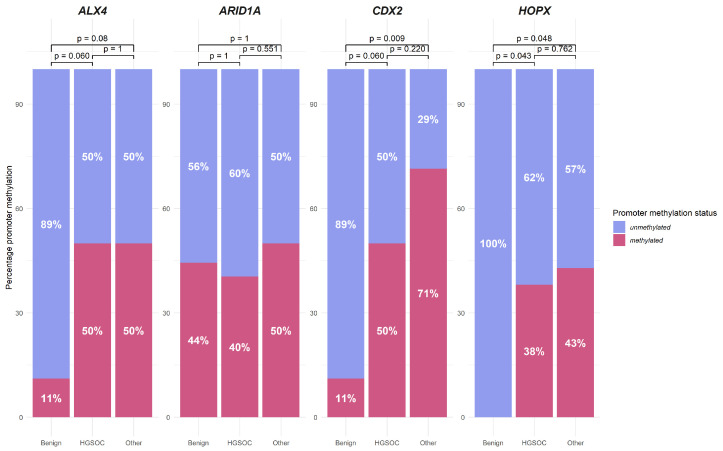
*HOX*-related (*ALX4*, *HOPX*, *CDX2*) and *ARID1A* gene promoter methylation in high-grade ovarian cancer (HGSOC, n = 42), non-HGOSC ovarian cancer (Other, n = 14) and benign (n = 9) cases.

**Figure 3 biomedicines-13-00441-f003:**
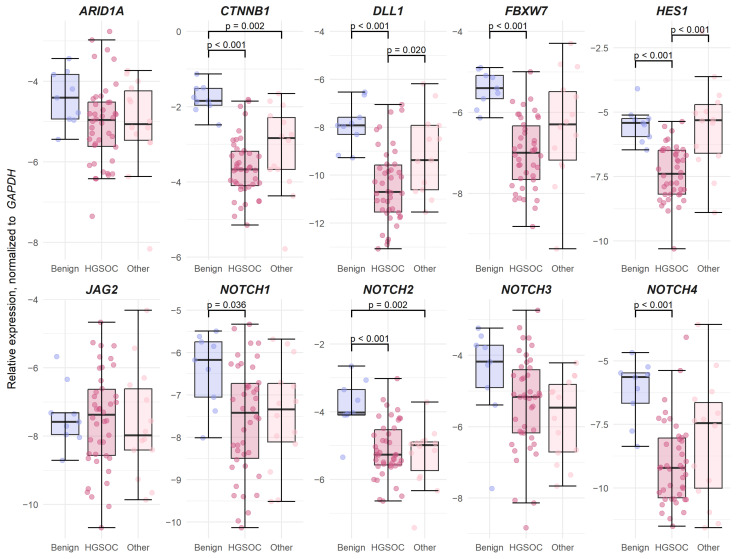
Boxplots depicting gene expression in three gynecologic tumor groups: benign (n = 9, blue), high-grade serous ovarian cancer (HGSOC, n = 42, dark pink), and non-HGSOC gynecologic tumors (Other, n = 14, light pink).

**Figure 4 biomedicines-13-00441-f004:**
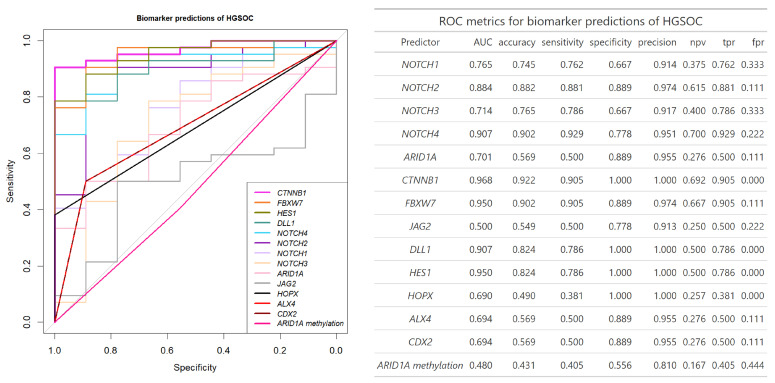
ROC plot and its metrics for the gene expression and promoter methylation biomarkers separation of the HGSOC (n = 42) and benign cases (n = 9). *ALX4* and *CDX2* gene promoter methylation biomarker curves overlap. AUC—area under the curve, npv—negative predictive value, tpr—true positive rate, fpr—false positive rate.

**Figure 5 biomedicines-13-00441-f005:**
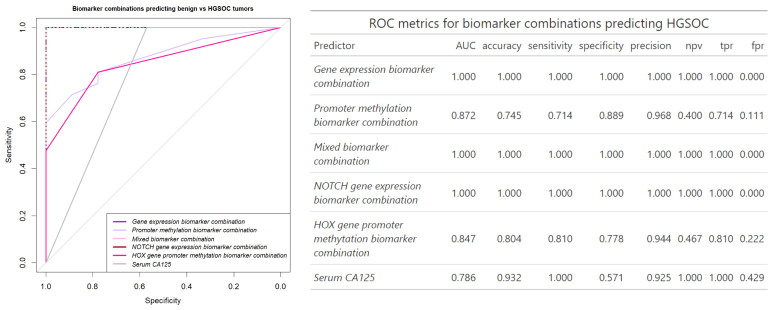
Receiver operator characteristic (ROC) plot and its metrics for the gene expression, promoter methylation biomarker, and serum CA125 biomarker separation of the HGSOC (n = 42) and benign cases (n = 9). Gene expression biomarker combination (all 10 genes), Mixed biomarker combination (all 14 biomarkers), and Notch gene expression biomarker combination (*NOTCH1-4*, *HES1*, *DLL1*, *JAG2*) curves overlap. *HOX* gene promoter methylation biomarker combination includes three genes (*HOPX*, *ALX4*, *CDX2*), serum CA125 threshold concentration considered 35 U/mL.

**Table 1 biomedicines-13-00441-t001:** Clinical features of the study cohort. NA—data not available; SD—standard deviation.

Clinical Feature	Ovarian Cancer	Benign Gynecologic Tumors	All
n	56	9	65
Histology group			
HGSOC	42 (75.00%)		42 (64.62%)
Other gynecologic malignant tumors	14 (25.00%)		14 (21.54%)
Benign gynecologic tumors		9 (100.00%)	9 (13.85%)
Average age at diagnosis (±SD)	59.16 (±9.69)	53.67 (±9.33)	58.40 (±9.76)
Median CA125 U/mL (min–max)	478.50 (34.00–5384.00)	18.22 (10.00–118.00)	419.30 (10.00–5384.00)
FIGO stage			
I	9 (16.07%)		9 (13.85%)
II	3 (5.36%)		3 (4.62%)
III	30 (53.57 %)		29 (44.62%)
IV	14 (25.00%)		14 (21.54%)
NA		9 (100.00%)	9 (13.85%)
Grade			
G1	6 (10.71%)		7 (10.61%)
G3	42 (75.00%)		42 (63.64%)
NA	8 (14.29%)	9 (100.00%)	17 (25.76%)

## Data Availability

The datasets used and analyzed during the current study are available from the corresponding author on reasonable request due to patient data privacy concerns.

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
