# Peer review of "Integrative Analysis of Gene Expression and Promoter Methylation to Differentiate High-Grade Serous Ovarian Cancer from Benign Tumors"

_biomedicines, 2025, doi:10.3390/biomedicines13020441_

Round 1

Reviewer 1 Report

Comments and Suggestions for Authors

This study assessed the methylation level of 10 target genes in 65 clinical samples, including HGSOC, non-HGSOC, and benign samples. These data could support future studies in targeting ovarian cancer cells with methylation regulating molecules. However, the manuscript need to be improved by incorporating a few changes. Here are some detailed comments: 

In the introduction section, the author should discuss previous studies and findings in methylation levels in ovarian cancer, as well as clinical trials that used molecules that regulate methylation levels (e.g., DNA methyltransferase inhibitor). The current introduction section is very general and does not focus on methylation in ovarian cancer. 

The study examined 14 biomarkers, the methylation level of 4 promoters and expresison of 10 target genes - 9 genes of Notch and Wnt, and ARID1A. Although the genes are important and some previous studies may be associated, how were these 14 biomarkers picked vs. other genes? It was unclear why the authors chose to examine these genes and why these genes are considered critical. The process of screening, identifying, and selecting these target genes should be described and rationales should be provided. 

The authors need to address the significance and impact of this study. How will this panel of 4 promoters and 10 genes be better than other current early detection biomarkers? Will it show any correlation with prognosis and outcome? These are not covered in this study; and as an article to report a new set of biomarkers for diagnosis, these questions should be answered and should be within the scope of this study.

All the figures are analyzing based on comparing HGSOC vs. non-HGSOC / benign samples; how were the methylation levels correlated with the stages of HGSOCs? Is it able to predict the prognosis level? 

Author Response

Thank You for Your time and kind and insightful comments. We made sure to improve the manuscript according to them.

  1. Reviewer comment:
    “In the introduction section, the author should discuss previous studies and findings in methylation levels in ovarian cancer, as well as clinical trials that used molecules that regulate methylation levels (e.g., DNA methyltransferase inhibitor). The current introduction section is very general and does not focus on methylation in ovarian cancer.
    The study examined 14 biomarkers, the methylation level of 4 promoters and expresison of 10 target genes - 9 genes of Notch and Wnt, and ARID1A. Although the genes are important and some previous studies may be associated, how were these 14 biomarkers picked vs. other genes? It was unclear why the authors chose to examine these genes and why these genes are considered critical. The process of screening, identifying, and selecting these target genes should be described and rationales should be provided.”

Thank you for the comments. We agree that the introduction could use some more in-depth discussion on the biomarkers chosen and your comments have lead the decision to re-write a part of introduction. We have expanded the introduction section to include some background on most common OC biomarkers including Notch and Wnt pathways (thus deciding to analyze their key genes), as well as background on epigenetic regulation, specifically ARID1A and HOX gene hypermethylation in other diseases as well as their use as targets for novel therapies. (Lines 41-77)

  1. Reviewer comment: “The authors need to address the significance and impact of this study. How will this panel of 4 promoters and 10 genes be better than other current early detection biomarkers? Will it show any correlation with prognosis and outcome? These are not covered in this study; and as an article to report a new set of biomarkers for diagnosis, these questions should be answered and should be within the scope of this study.”

Unfortunately, we do not have reliable data on the prognosis and outcomes, although it would be beneficial to look at it. However, we made sure to expand the discussion section to highlight the biomarkers that were correlated with clinical features and made a more in-depth comparison with the available biomarker CA125 to show the significance of study and selected biomarkers. (Lines 356-377)

  1. Reviewer comment: “All the figures are analyzing based on comparing HGSOC vs. non-HGSOC / benign samples; how were the methylation levels correlated with the stages of HGSOCs? Is it able to predict the prognosis level?”

We only included the methylation figures depicting statistically significant results. However, your comment made us re-examine the results and we have missed that in the HGSOC group, ARID1A methylation has shown significant association with FIGO stage with no methylated cases found in stage II cases p = 0.048. Thus, we have expanded the supplementary materials to include this result (Supplementary Figure 2) as well as the text (lines 206-208). However, all other associations with clinical features remained non-clinically significant, thus are not shown in separate figures.

Reviewer 2 Report

Comments and Suggestions for Authors

Dear Authors,

Thank you for submitting your manuscript titled "Integrative Analysis of Gene Expression and Promoter Methylation to Differentiate High-Grade Serous Ovarian Cancer from Benign Tumors" to our journal. This work addresses a critical topic in ovarian cancer research, and I commend your clear structure and comprehensive discussion. However, to enhance the clarity and depth of your manuscript, I recommend the following revisions:

Could you consider analyzing additional genes? Incorporating more biomarkers might yield more precise results and provide a broader understanding of the genetic underpinnings of high-grade serous ovarian cancer. The introduction could also be expanded to provide more context for the choice of genes like ARID1A and the databases selected. Discussing the advantages of your approach and whether these genes are also highly expressed in other types of cancer could clarify the broader applicability of your findings.

In the methods section, please clarify whether ethical approval was obtained for using patient samples, including the ethic number. Additionally, providing a rationale for your choice of DNA or RNA extraction methods and cDNA synthesis would be beneficial. You might consider citing relevant literature, such as https://doi.org/10.3390/biom13050744 and J Vis Exp. 2009 Aug 7;(30):1470. doi: 10.3791/1470, which could lend more credibility and depth to your methodology.

Regarding your statistical analysis, why was an unpaired T-test not chosen for the two independent sample analyses? A discussion on the selection of statistical tests could enhance the reader's understanding of the data analysis strategies used. The discussion could benefit from addressing future challenges in ovarian cancer research more broadly. For example, exploring the implications of "negative-biomarkers" ovarian cancer types, which do not express typical genetic markers, would provide valuable insights into the complexities of this disease.

Regarding Figure 1, the font size is too small for easy readability, and some text appears distorted. Resizing the text within the figure to ensure it is accessible to all readers is essential. More explicitly articulate the advantages of the algorithm used over existing methods. A detailed comparison highlighting improvements in accuracy, efficiency, or applicability to different datasets would better showcase the innovation of your work and its practical implications in medical research and treatment strategies.

Addressing these points will not only clarify and enhance your manuscript but also emphasize its contribution and potential impact on the field of ovarian cancer research. I look forward to your revisions.

Author Response

Most sincere Thank you for Your time and perceptive comments. We made sure to improve the manuscript according to them.

  1. Reviewer comment: “Could you consider analyzing additional genes? Incorporating more biomarkers might yield more precise results and provide a broader understanding of the genetic underpinnings of high-grade serous ovarian cancer.”

Thank you for your kind comments. Currently the addition of more genes is beyond the scope of this manuscript due to time and funding constraints. However, we are working on selecting additional target genes for future studies.  

  1. Reviewer comment: “The introduction could also be expanded to provide more context for the choice of genes like ARID1A and the databases selected. Discussing the advantages of your approach and whether these genes are also highly expressed in other types of cancer could clarify the broader applicability of your findings.”

We agree that the introduction could have used a more in-depth discussion on the biomarkers, thus we have since rewritten a part of the introduction to include more background on the chosen biomarkers and their expression in other diseases. The main database used for the biomarker selection was the TCGA though papers analyzing its results, which we have cited in the introduction (lines 41-77).

  1. Reviewer comment: “In the methods section, please clarify whether ethical approval was obtained for using patient samples, including the ethic number.”

The approval number was provided in the “Institutional review” section at the end of the manuscript; however, it is now provided in the methods as well. (Lines 104-105)

  1. Reviewer comment: “Additionally, providing a rationale for your choice of DNA or RNA extraction methods and cDNA synthesis would be beneficial. You might consider citing relevant literature, such as https://doi.org/10.3390/biom13050744 and J Vis Exp. 2009 Aug 7;(30):1470. doi: 10.3791/1470, which could lend more credibility and depth to your methodology.”

Both the DNA and RNA extraction were performed using the current gold-standard methods vastly validated in many types of samples. We are grateful for the links provided, as we have found the second article especially relevant. We added rationale and relevant citations to the nucleic extraction section. (Lines 120-131)

The cDNA and qPCR kits choice has been informed by the availability and cost-efficiency. However, relevant studies have found the cDNA kit to be comparable to other cDNA synthesis kits on the market and generate sufficient cDNA yield for PCR applications. We have included this information in the manuscript (line 141). Additionally, we can personally vouch for the chosen method as the cDNA synthesis kit was created by one of our co-authors.

  1. Reviewer comment: “Regarding your statistical analysis, why was an unpaired T-test not chosen for the two independent sample analyses? A discussion on the selection of statistical tests could enhance the reader's understanding of the data analysis strategies used.”

Given the unequal sample sizes in our study (56 malignant gynecologic tumors vs 9 benign gynecologic conditions), Welch’s t-test was the preferred approach to minimize the risk of invalid statistical inferences. It has also been reported that Welch’s t-test improves statistical power, and its adjustment using the Benjamini and Hochberg method effectively controls false discovery rate (Type I errors). Considering these assumptions, we believe Welch’s t-test provided a more conservative method for comparing the two independent groups. We have included a condensed explanation in the statistical analysis section of the methods alongside relevant citations (lines 174-177).

  1. Reviewer comment: “The discussion could benefit from addressing future challenges in ovarian cancer research more broadly. For example, exploring the implications of "negative-biomarkers" ovarian cancer types, which do not express typical genetic markers, would provide valuable insights into the complexities of this disease.”

Thank you for the comment. We have expanded the discussion section to highlight current challenges of ovarian cancer research and highlighted the need for new biomarkers (350-377).

  1. Reviewer comment: “Regarding Figure 1, the font size is too small for easy readability, and some text appears distorted. Resizing the text within the figure to ensure it is accessible to all readers is essential.”

Thank you for raising the issue, we have resized and increased the resolution of the figure as much as we could and hope that the image is now improved in legibility (Figure 1, line 196).

  1. Reviewer comment: “More explicitly articulate the advantages of the algorithm used over existing methods. A detailed comparison highlighting improvements in accuracy, efficiency, or applicability to different datasets would better showcase the innovation of your work and its practical implications in medical research and treatment strategies.”

Thank you for the comment. We have since expanded the discussion to highlight the improvements of the multiple biomarker panel over the use of single biomarkers. Although the use of single biomarker tests (e.g. CA125) is cost-effective, it is unlikely that single biomarkers could achieve the same diagnostic power as multiple biomarkers, due to heterogeneity of the OC, which we now commented on in the discussion. We have provided a more detailed explanation of the multiple gene expression biomarker model improvement over CA125 and highest AUC single biomarker (CTNNB1 expression) as well (367-385). 

Round 2

Reviewer 1 Report

Comments and Suggestions for Authors

The authors have addressed my questions; I do not have further comments.